

**Vegetation and fire anomalies during the last ~70 ka in the Ili Basin, Central Asia, and their**
**implications for the ecology change caused by human activities**
**Yunfa Miao [a, b]\*, Yougui Song [b, c]\*, Yue Li [b, d], Shengli Yang [e], Yun Li [b]**
a. Key Laboratory of Desert and Desertification, Northwest Institute of Eco-Environment and
Resources, Chinese Academy of Sciences, Lanzhou 730000, China
b. State Key Laboratory of Loess and Quaternary Geology, Institute of Earth Environment,
Chinese Academy of Sciences, Xi'an 710061, China
c. Research Center for Ecology and Environment of Central Asia, Chinese Academy of Sciences,
Urumqi, 830011, China
d. College of Resources and Environment, University of Chinese Academy of Sciences, Beijing,
100049 China
e. Key Laboratory of Western China's Environmental Systems(Ministry of Education), College of
Earth and Environmental Sciences, Lanzhou University, Lanzhou, 730000, China
Corresponding author: miaoyunfa@lzb.ac.cn; ygsong@loess.llqg.ac.cn
**Abstract:** Changes in vegetation characteristics and fire occurrence during the last glacial period
offer an opportunity to better understand paleoclimate change and past human activities as well as
the relationships among them. However, in central Asia, records of both vegetation and fire have
rarely been obtained from the same profile. Here, for the first time, we present pollen and
microcharcoal data collected together from the wind-blown loess Nileke section, representing the
past ~70 thousand years (ka) in the Ili Basin, Northwest China, Central Asia. These records enable
investigation of the pollen-based vegetation and microcharcoal-based fire proxies as well as their
possible relationships with ancient human activities. The results show that the temperate
herbaceous taxa remained at relatively low levels before 36 ka, whereas the temperate woody taxa,
especially Cupressaceae, were abundant. At the same time, the fire frequencies were relatively low.
After 36 ka, herbaceous taxa abruptly replaced Cupressaceae and the fire occurrence gradually
increased. We named this change as the local vegetation degeneration event, because no
equivalent changes have been identified anywhere else across Eurasia. Prior to the event, a period

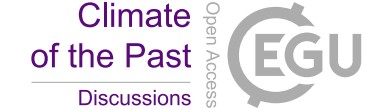

of intensified fire activity occurred between 47.5 and 36 ka, although the background fire activity
was relatively low. We argue that the intensified local fire activity was the primary factor causing
the vegetation event and was mainly driven by human activity. Following migrations from Africa
after 200 ka, humans began to colonize the Ili Basin at least 47.5 ka ago, bringing their skills of
fire control and consequential destruction of woody vegetation. Future analysis of first-hand
archeological sites in this area will be an important step in supporting our hypothesis.

**Keywords**: Vegetation; Fire; Ecology; Human activities; Last glacial period

**1. Introduction**

The climate, vegetation, fire and human activities, as well as the relationships among them

over the late Quaternary, especially the last glacial period, provide basic insights by which to
understand the future (e, g., Behling and Safford, 2010; Cheng et al., 2012; Li et al., 2013; Hubau
et al., 2015; Varela et al., 2015). High-resolution stalagmite (Wang et al., 2001; Cheng et al., 2016),
ice core (Thompson et al., 1997; Petit et al., 1999; Augustin et al., 2004) and loess (e.g., Chen et
al., 1997; Hao et al., 2012; Sun et al., 2012) analysis has yielded highly reliable, integrated
paleoclimate records. These are characterized by a series of strong fluctuations, named cold
Heinrich or warm Dansgaard-Oeschger events, as well as a warm middle Holocene (e.g., Bond et
al., 1997). At the eastern margin of Central Asia, precipitation has followed the same patterns as
these events: lower precipitation during the cold events and vice versa (e.g., Rao et al., 2013).
Vegetation is regarded as one of the most sensitive proxies for climate change, and a limited
number of complete vegetation records have been obtained to show how the terrestrial ecological
landscape responded to climate change (e.g., Guiot et al., 1993; Allen et al., 1999; Jiang et al.,
2011). Fire is another sensitive proxy used for reconstructing climate (e.g., Filion, 1984; Bird and
Cali, 1998; Bowman et al., 2009). Besides climate, records of vegetation and fire are also unique
indicators of human activities, owing to the impact of human activities such as vegetation cutting
and burning (e.g., Patterson et al., 1987; Whitlock and Larsen, 2002; Huang et al., 2006; Aranbarri
et al., 2014; Miao et al., 2016b; Sirocko et al., 2016); however, most relevant studies have been
limited to the late Holocene, especially at archeological sites. Few studies have attempted to
reconstruct the last glacial period, despite this period being considered as a key period of



migration: the human migration from Africa started at around 200 ka (Templeton, 2002; Sun et al.,
2012). Furthermore, studies of vegetation and fire within the same profile (section or core) are
helpful in understanding the vegetation, fire and climate, as well as human activities (e.g., Zhao et
al., 2010; Xiao et al., 2013; Miao et al., 2016a; b).

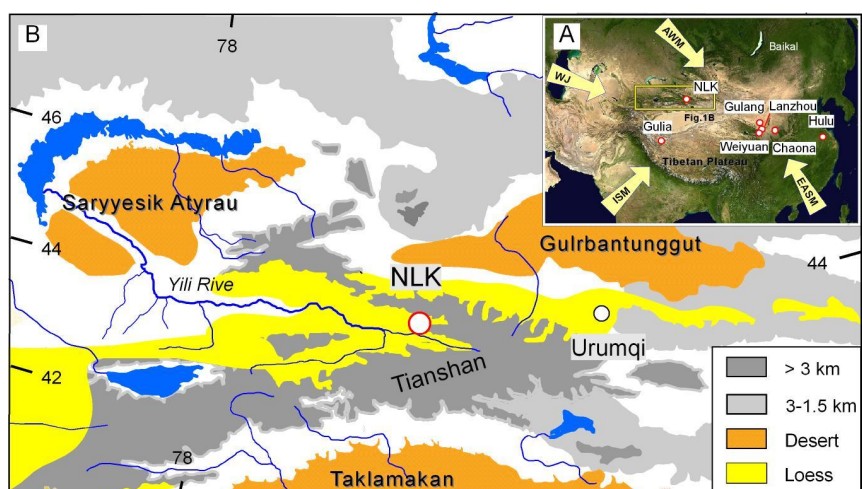


*Figure 1. A. Asian morphological map with climate systems showing the NLK (Nileke) section*

*location and climatic proxy sites covering the past 70 ka. These sites include the Gulia glacial*

*core (Thompson et al., 1997), Gulang wind-blown sediments (Sun et al., 2012), Chaona (Wang et*

*al., 2016), Hulu stalagmite oxygen isotope records (Wang et al., 2001), Weiyuan summer*

*precipitation reconstruction (Rao et al., 2013) and Lanzhou pollen analysis (Jiang et al., 2011). B.*

*A morphological map showing the location of the Nileke section in this study.*


Central Asia is dominated by a dry climate (Figure 1A), which is very sensitive to any
climate changes (fluctuations or abnormality) and human activities. In this study, we firstly
present pollen and microcharcoal results from a wind-blown loess sediment section (Figure 1B) to
reveal how vegetation and fire activity have changed during the past 70 ka; we then analyze the
mechanisms underlying these changes.
**2. Materials and methods**
**2. 1 Lithostratigraphy and chronology**
The Ili Basin is surrounded by the Tianshan orogenic belt in east Central Asia, with gentle
topography to the west. The basin opens to the west and funnels winds and cyclonic disturbances,

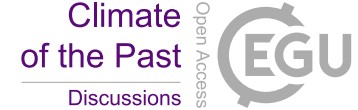



often associated with prevailing westerly winds, down its axis (Ye, 2001). The Ili Basin has a
temperate, continental, arid climate with a mean annual temperature that varies from 2.6 ℃ at
1850 m to 10.4 ℃ at 660 m; the mean annual precipitation varies correspondingly from 512 to 257
mm (Ye et al., 1997; Ye, 2001). The surface soils are a sierozem (aridosols) with widely
distributed desert steppe vegetation. The vegetation coverage is <50%, mainly comprising
*Artemisia* spp. and Chenopodiaceae spp. (Ye et al., 2000). There are no obvious accumulations of
organic matter in the surface horizon of the modern soil.

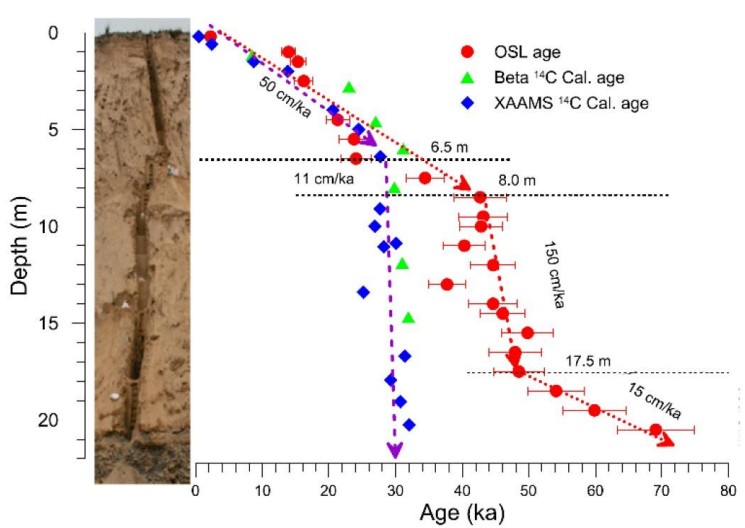


*Figure 2. Stratigraphy and dating for the Nilke Section (for more detail see Song et al. 2015).*

To the west of the Ili Basin are the vast central Asian Gobi Deserts, such as Saryesik-Atyrau
Desert (Figure 1B), the probable source of dust for Late Pleistocene loess deposits. The loess
deposits are widely distributed across the piedmont of the Tianshan Mountains, river terraces and
desert margins. The loess thickness ranges from several meters to approximately two hundred
meters, and there are two primary depocenters: around Sangongxiang in the northwest and
Xinyuan in the east Ili basin (Song et al., 2014). Most of the loess appears to have been deposited
since the last interglacial period (ca. 130 ka; Ye, 2001; Song et al., 2010; 2014; Li et al., 2016).
The Nileke section (83.25 °E, 43.76 °N, 1253 m a. s. l) is located on the second terrace of the
Kashi River, a branch of the Ili River, in the east Ili Basin (Figure 1B). The loess sequence is 20.5





m thick, largely homogeneous in appearance with two diffuse paleosols at depths of 5-7.5 m and
15.5-18.5 m identified by the extent of rubification (Figure 2) (Song et al., 2015). The loess
sequence rests conformably on fluvial sand and gravels. The contact between the loess and fluvial
sediment is abrupt with no obvious lag, erosion or pedogenesis. The loess is composed of 70%-84%
silt and 3%-17% very fine sand (63-100 mm), with the remaining fraction being clay. A
high-resolution quartz optically stimulated luminescence (OSL) chronology has already been
established (Yang et al., 2014; Song et al., 2015). Based on OSL ages, two intervals of higher mass
accumulation rate occurred at 49-43 ka and 24-14 ka (Song et al., 2015).
**2.2 Pollen and charcoal collection**
A total of 104 samples of 49-56 g weight were taken at 20 cm intervals from the Nileke
section for palynological analysis. The samples were treated with standard palynological methods:
acid digestion (treatment with 10% HCl and 40% HF acid to remove carbonates and silicates,
respectively) (Li et al., 1995) and fine sieving to enrich the spores and pollen grains. The prepared
specimens were mounted in glycerol for identification. All samples were studied at the Cold and
Arid Regions Environmental and Engineering Research Institute (CAREERI), Chinese Academy
of Sciences (CAS), by comparison with official published pollen plates and modern pollen
references. Each pollen sample was counted under a light microscope at 400× magnification in
regularly spaced traverses. More than 150 spores and pollen grains were counted within each
sample. A known number of *Lycopodium clavatum* spores (batch # 27600) were initially added to
each sample for calculation of pollen and microcharcoal concentrations (Maher, 1981).
The concentration of pollen or microcharcoals can be calculated according to the following
formula: $C = N_x / L_x \times 27600 / W_x$
C: concentration; $N$: identified number of charcoals; $L$: number of *Lycopodium clavatum*; $W$:
sample dry weight; x: sample number; 27600: grain numbers of *Lycopodium clavatum* per pill.
For the microcharcoal identification, four particle size units were defined as follows: <30 μm,
30-50 μm, 50-100 μm and >100 μm (Miao et al., 2016a), then the total microcharcoal
concentrations (MC) were obtained by summing over all sizes and using the above formula. As
the residual matter from the incomplete burning of vegetation, charcoals are usually characterized
by either spherical bodies without structure or particles with some original plant structures
preserved.

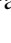

**3. Results and analysis**
In the pollen assemblages, dominant palynomorphs originated mainly from herbaceous taxa
such as Chenopodiacaee, *Artemisia*, Ranunculaceae, Asteraceae and Rosaceae. Woody taxa were
Cupressaceae, *Pinus*, *Betula*, Ulmaceae and Tamaricaeae; the other temperate taxa with low
percentages were *Quercus*, *Picea*, *Cedrus* and *Broussanetia* etc.

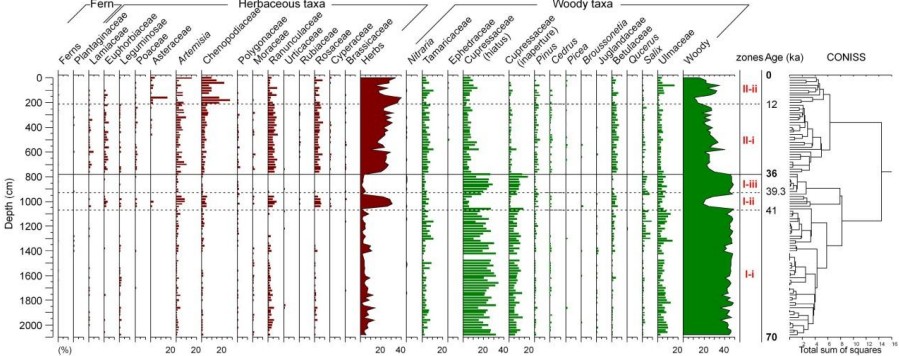

*Figure 3. Pollen percentage diagram for the Nileke section, Ili Basin.*

The pollen diagram was divided into two pollen assemblage zones based on variations in the
percentages according to stratigraphically-constrained cluster analysis (CONISS) carried out using
Tilia software (E. Grimm of Illinois State Museum, Springfield, Illinois, USA) (Figure 3) and
concentrations of the dominant taxa, from the older to the younger samples. The two zones are as
follows.
Zone I (2080-780 cm; 70-36 ka): the assemblages were characterized by high percentages of
Cupressaceae (hiatus) (ca. 5.2%-68.7%, with an average of 42.4%) and Cupressaceae (inaperture)
(ca. 1.4%-34.7%, average 14.0%), Ulmaceae (ca. 2.8%-26.1%, average 11.3%), Tamaricaceae (ca.
1.9%-20.9%, average 7.3%). In the herbaceous taxa, only *Artemisia* (ca. 0-14.8%, average 3.3%),
Rannuculaceae (ca. 0-14.2%, average 3.0%) and Chenopodiacae (ca. 0-8%, average 1.8%) were
dominant, and at much lower abundances relative to the woody taxa. In more detail, three
subzones were identified according to the assemblages: I-i, I-ii and I-iii with divisions at 1070 and
930 cm, corresponding to ages of 41 ka and 39.3 ka. The subzones I-i and I-iii were both
characterized by high Cupressaceae, whereas subzone I-ii was relatively dominated by herbaceous



taxa.

In the pollen concentrations, the same zones were also identified at a depth of 780 cm. The

woody taxa were dominant below this boundary, and those such as Cupressaceae (hiatus and
inaperture), Ulmaceae and Tamaricaceae reached counts of around 1000 grains/g, 200 grains/g
and 100 gains/g respectively. Others such as *Pinus*, Juglandaceae, *Betula* and *Salix* were also
common. By contrast, all herbaceous taxa were very low (Figure 4). We also added the boundary
at a depth of 780 cm to divide the MC assemblages. Below the boundary, the fluctuations in all
different sizes and shapes were stronger, especially in Zones I-ii and I-iii (Figure 5).

Zone II (780-0 cm; 36-0 ka): the woody taxa were extensively replaced by herbaceous taxa,

of which Cupressaceae (hiatus) (ca. 3.5%-51.0%, average 12.1%) and Cupressaceae (inaperture)
(ca. 0-24.5%, average 2.9%), Tamaricaceae (ca. 1.5%-19.4%, average 8.9%) and Ulmaceae (ca.
0.5%-27.9%, average 5.6%) were dominant; *Betula* and *Pinus* increased slightly (ca. 0-12.6%,
average 6.4% and ca. 0-8.6%, average 2.3%, respectively). In the herbaceous taxa, *Artemisia* (ca.
0.9-24.1%, average 7.1%), Chenopodiaceae (ca. 0-48.2%, average 9.0%), Rosaceae (ca. 0-15.0%,
average 8.6%) and Rannuculaceae (ca. 0-14.2%, average 3.0%) increased obviously, and the rest
remained broadly stable. In more detail, two sub-horizons were identified: II-i and II-ii, divided
based on the Asteraceae and Chenopodiaceae increase at 210 cm, correlated to an age of 12 ka B.P.
(Figure 3).

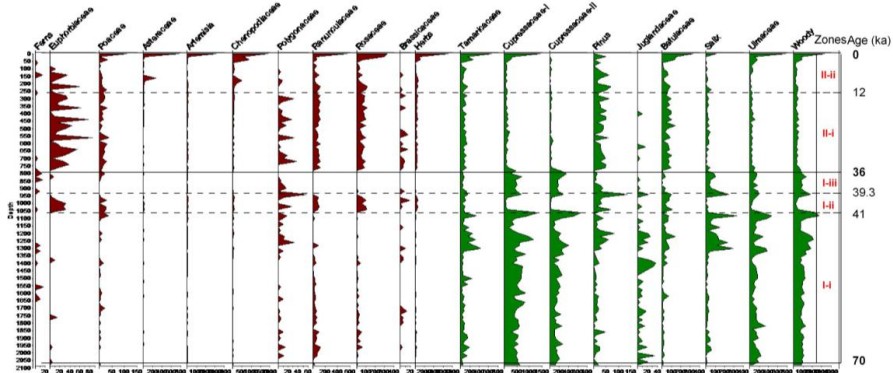

*Figure 4. Pollen concentration diagram for the Nileke section, Ili Basin, China (zone divisions*
*follow Figure 3).*

The pollen concentrations in Zone II show that the woody Cupressaceae (hiatus and

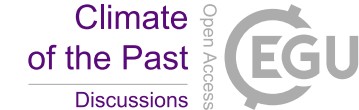



inaperture), Ulmaceae, Juglandaceae and Tamaricaceae decreased obviously while the herbaceous
taxa such as *Artemisia*, Chenopodiaceae, Poaceae, Ranuculaceae and Rosaceae increased. At the
sub-boundary of II-i and II-ii, Asteracea, *Artemisia* and Chenopodiaceae increased strongly
(Figure 4). For the MC, all different shapes and sizes remained at generally stable and relatively
low values in Zone II-i whereas in Zone II-ii the concentrations in all samples clearly started to
increase, especially in the uppermost layers (Figure 5).
In summary, there are clear divisions at a depth of 780 cm, corresponding to an age of 36 ka.
Prior to this change there was a high percentage of woody taxa, but subsequently the herbaceous
taxa became more dominant, especially after 12 ka. The assemblages of pollen concentrations and
MC can also be divided into two periods, with a transition at 36 ka.

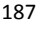

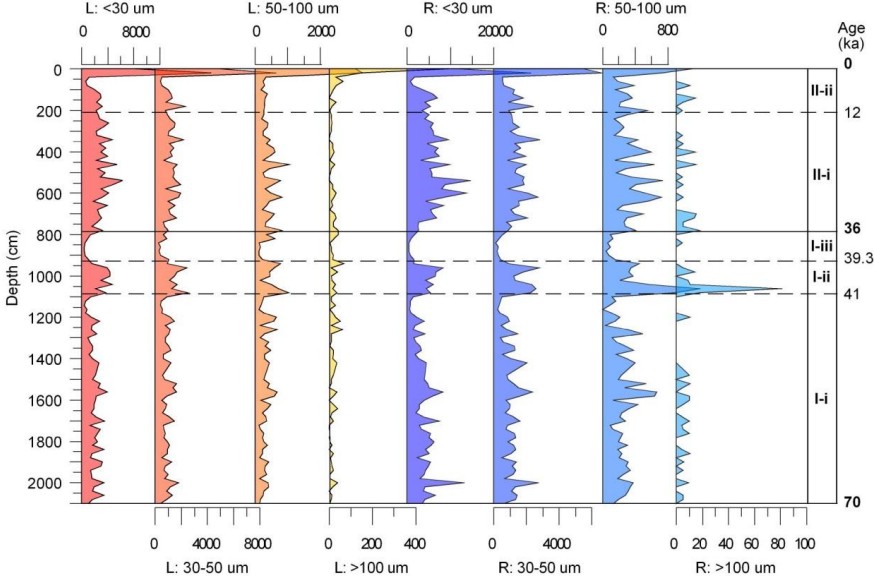


*Figure 5. The MC records for different sizes and shapes in the Nileke section (unit: grains/g; L:*
*elongated; R: zone divisions follow Figure 3).*

**4. Discussion**
The modern climate in Central Asia is controlled by the East Asian summer monsoon, Indian
summer monsoon, Asian winter monsoon and Westerlies (Figure 1A). In the Ili Basin,
meteorological records indicate that strong surface winds from the west, northwest and southwest





which occur frequently from April to July play the dominant role in the transportation of dust,
suggesting that the wind-blown sediments in the Nileke section are driven by the Westerlies.
Therefore, the grain size of the sediments can be regarded as a basic proxy for the intensification
of the Westerlies (Li et al., 2015; Li et al., 2016). Furthermore, the Ili Basin is surrounded by the
Tianshan Mountains to the south, east and north (with elevations exceeding 3-4 km) but low
elevations (~800-1600 m a. s. l) to the west. Consequently, most of the precipitation reaching the
basin will have been transported by the Westerlies during the last glacial period. Here, we try
firstly to estimate changes in the vegetation and fire characteristics in the Ili Basin, secondly to
discuss the overall climate change across Eurasia over the past 70 ka, and finally provide some
speculation regarding the differences.
**4.1 Vegetation and fire records at Nileke**
The pollen dataset can be regarded as a reliable proxy for investigating the vegetation change
in the study area. In the Nileke section, during 70-36 ka, the pollen assemblages show a relatively
woody taxa-dominated landscape: during this time, the woody taxa reached their highest levels of
the whole section (Figure 6). After 36 ka, the vegetation deteriorated markedly, as evidenced by
the rapid disappearance of woody taxa following strong fluctuations during 41-36 ka. This was
especially notable for Cupressaceae. In more detail, no obvious fluctuations were noted during
these two periods except for the interval between 41 and 36 ka. The pollen concentrations also
follow a similar trend, according to the pollen percentages. Overall, the most obvious vegetation
change according to the pollen data was at around 36 ka ago, as indicated by the sharp change in
vegetation assemblages. A similar transition has not been observed in Europe (e.g., Guiot et al.,
1993; Allen et al., 1999) or elsewhere in Asia (e.g., Jiang et al., 2011).
Charcoal particles remaining following combustion are entrained by the smoke and then
carried by the wind. Following deposition, they remain as a direct proxy of fire activity. On the
Loess Plateau, smaller charcoal particles can be easily transported over long distances by the wind,
but the larger particles tend to travel only a short distance (Huang et al., 2006). Therefore, the
charcoal particle size can be related to its distance from the fire (Patterson et al., 1987; Clark, 1988;
Luo et al., 2006; Miao et al., 2016a; b), with smaller particles likely to have been transported
further from the fire (Clark, 1988). Moreover, a rounder shape (long axis to short axis ratio <2.5)
is more likely related to forest fires while elongated particles (long axis to short axis ratio >2.5)

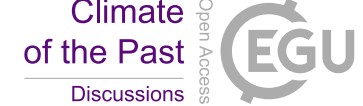



are more indicative of grass fires (Umbanhowar and Mcgrath, 1998; Crawford and Belcher, 2014).
The charcoal assemblages in the Ili Basin show a relative low fire frequency/severity at regional
and local scales, in forest and grass, before 36 ka; activities then increased gradually after 36 ka
(Figure 6, 7). Superimposed on this general trend, the first notable anomaly occurred at 47.5-36 ka
and was characterized by a high frequency of local grass and forest fires. Another similar anomaly
occurred at the top of the profile (less than 6 ka) in the layer with the highest levels of regional and
local grass fires as well as the highest regional forest fires (Figure 3-5).

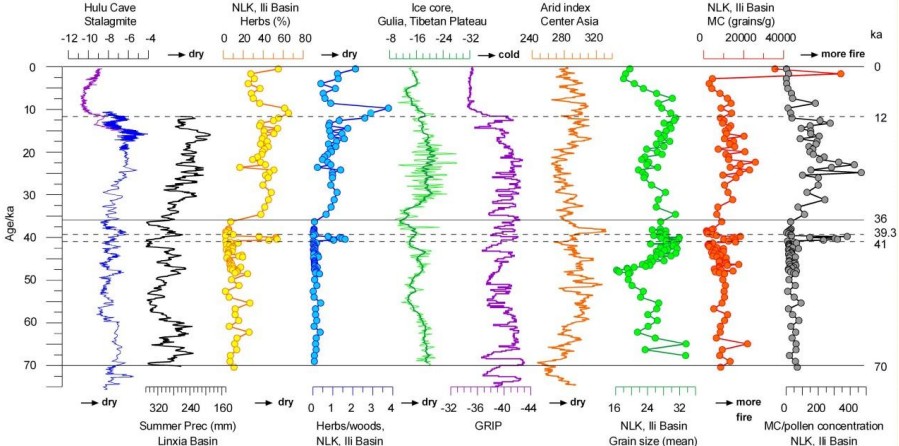


*Figure 6. Comparison of climate proxies across the Northern Hemisphere and Nileke section.*
*These are Hulu cave, Nanjing (Wang et al., 2001); summer precipitation reconstruction in the*
*Linxia Basin (Rao et al., 2013); ice core, Gulia, Tibetan Plateau (Thompson et al., 1997); NGRIP*
*(Andersen et al., 2004); and aridity index in central Asia (Li et al., 2013). Divisions follow Fig. 3.*
*No anomalies occurred during 41-36 ka.*

**4.2 Climate in Eurasia**
Here, multiple proxies from the terrestrial and marine sources have revealed the basic patterns
of climate change during the last glacial period, characterized by abrupt, millennial-scale cold
events (Petit et al., 1999; Wang et al., 2001; Augustin et al., 2004; Cheng et al., 2016) (Figure 6).
These climate fluctuations are particularly pronounced in records of the East Asian monsoon
system (Porter and An, 1995; Guo et al., 1996; Thompson et al., 1997; Wang et al., 2001; Sun et
al., 2012).



The Greenland NGRIP ice core (Andersen et al., 2004) indicates that temperature variations
in the high latitudes of the Northern Hemisphere are characterized by high-frequency fluctuations
over the past 70 ka, with the most obvious change occurring at around 12 ka and no significant
anomaly at 36 ka. At the same time, high-resolution summer precipitation variations in the
western Chinese Loess Plateau were found to contain similar anomalies (Rao et al., 2013), yet
with no obvious precipitation change at around 36 ka, despite their proximity to the Lanzhou loess
sediments, where the shrubs and herbs reached the highest abundances after ~40 ka owning to the
westerlies strengthening and supplying plenty of moisture to Northwest China (Jiang et al., 2011).
In Europe the newest study shows that during 49-36.5 ka, the boreal forest of pine, birch and few
spruce with little dust activity, however the charcoal indicates drought stress and frequent forest
fires. During 36.5-28.5 ka, the steppe with grass, pine and birch enlarged. Dust storm increased.
Spread of anatomically modern humans in the increasingly open landscape, where horse, reindeer
and mammoth, the favored hunting preys, must have been abundant (Sirocko et al., 2016). This
time is correlated with the time of early modern humans spreading into central Europe (Trinkaus
et al., 2003; Mellars, 2006; Conard and Bolus, 2008; Klein, 2008; Hublin, 2012; Nigst et al.,

2014).

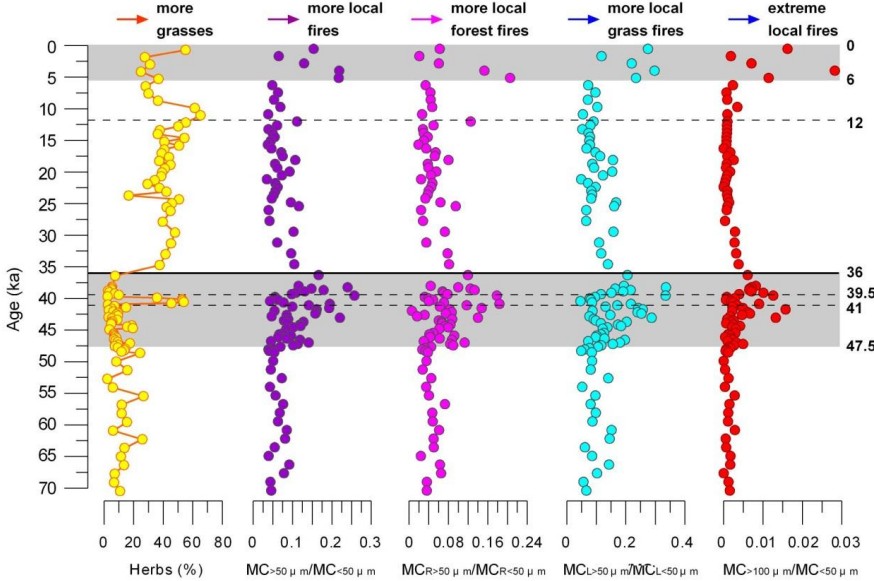


*Figure 7. Vegetation versus fire anomalies identified in the Nileke section during 47.5-36 kyr.*

*Gray rectangles show periods of intensified local fire activity during 47.5-36 and 6-0 kyr, which*



*cannot be explained as the result of the climate change.*

Therefore, we argue that the natural climate change at around 36 ka is not the main cause for
the vegetation changes in the Ili Basin. Furthermore, the aridity index in Central Asia reveals that
the change at ~36 ka did not shift the climate away from its generally arid classification (Li et al.,
2013). Another potential factor to consider is the wind velocity change, however according to the
grain-size distribution of the sediments in the Nileke section, there was no obvious change in the
mean size and accordingly no significant variation in wind during that time (Figure 6).
**4.3 Climate and fire anomalies and their driving forces**
According to the oxygen isotope records from Greenland (Andersen et al., 2004) and Hulu
Cave (Wang et al., 2001), as well as data from summer precipitation (Rao et al., 2013) and the
aridity index established for Central Asia (Sun et al., 2012), the climate across Central Asia has
maintained steady large-scale patterns with no substantial changes since 36 ka. Levels of $CH_4$
(Blunier and Brook, 2001) and $CO_2$ (Ahn and Brook, 2008) during this period remained within the
bounds of normal fluctuations. So, large-scale climate change across Eurasia cannot be the
primary factor explaining vegetation anomalies in the Ili Basin.
Excluding climate change, fire can be another factor causing changes to vegetation and land
cover (Miao et al., 2016a), with potential for then causing a climate anomaly. In Figure 7, we
compiled the microcharcoal data to investigate the fire intensity on a relatively regional scale
($MC_{>50 \ \mu m}/MC_{>50 \ \mu m}$), including local forest fire ($MC_{R>50 \ \mu m}/MC_{R>50 \ \mu m}$) and local grass fire
($MC_{L>50 \ \mu m}/MC_{L>50 \ \mu m}$) as well as extreme local fire events ($MC_{>100 \ \mu m}/MC_{<50 \ \mu m}$), according to the
different shapes and sizes (see section 4.1). The results revealed two obvious fire anomaly periods:
one during 47.5-36 ka, when local and extreme-local fires were markedly more intense, with a
sharp decrease at 36 ka; the second was during 6-0 ka, again characterized by strong local and
extreme-local fires.
In nature, wildfire has existed since the vegetation began to colonize the land (Glasspool et
al., 2004). According to Holocene fire records from the Northeast Tibetan Plateau (Miao et al.,
2016b), as well as global records on orbital time scales (Bird and Cali, 1998; Luo et al., 2001), the
climate change might have strongly driven the fire changes by changing humidity. Summer
precipitation during 41-36 ka was at its highest level of the past 70 ka (Rao et al., 2013), which

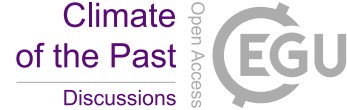



will have impeded burning. So, the precipitation change was not the key factor in the observed fire
anomalies. Another possibility is that the fire was caused by human activities. The earliest
human-controlled fire can be traced back to at least 0.8 million years in Israel (Goren-Inbar et al.,
2004) or 0.4-0.5 million years for *Homo erectus pekinensis* in China (Weiner et al., 1998), which
means that after that the humans have colonized the worldwide regions in the latest period of the
Pleistocene e.g., the last glacial period with the skills of fire control. The Ili Basin, as one of most
important passageways from Africa to high-latitude of Asia, e.g., Baikal Lake, can be burned
during their colonization, thus the natural vegetation during their colonization should have been
changed or destroyed strongly, especially including the arbors. Cupressaceae as one sensitive
woody species in the mid latitude of Inner Asia grow slowly and, once destroyed, regrowth is very
slow. This could explain why Cupressaceae disappeared so fast following human colonization.

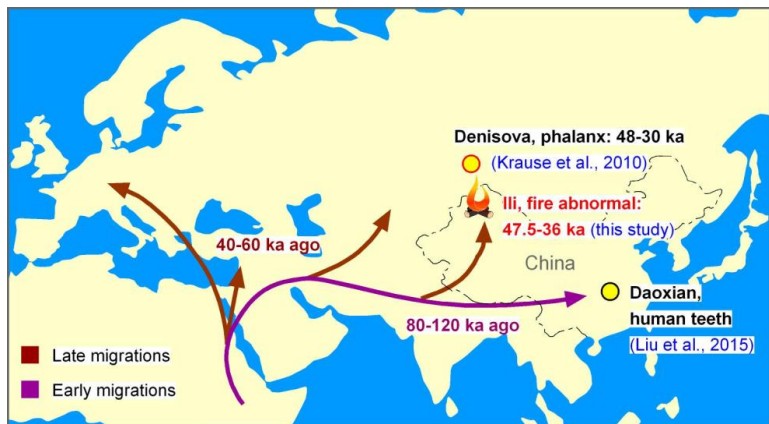

*Figure 8. An early migration from Africa (adapted from Callaway, 2015). Finds in the Ili Basin dated*
*to 47.5-36 kyr correlate with human fire activity (this study).*


There is widespread evidence supporting human occupation of Central Asia during the
Holocene (Huang et al., 1988; Wang and Zhang, 1988; Taklimakan Desert archaeology group,
1990; Yidilis, 1993; Lü et al., 2010; Zhang et al., 2011; Tang et al., 2013; Han et al., 2014). In the
Ili Basin, although direct archeological sites are limited, the coeval local fire intensification
supports human activity as a factor causing fire anomalies after around 6 ka. This relationship can
be similarly extended to observed fire anomalies at 47.5-36 ka, when humans migrated into the Ili





Basin. Although direct archeological proofs of fire usage at this time are still lacking, human
colonization of mid-to high-latitude Eurasia occurred after 200 to 80 ka (Wu et al., 2015) and
extended to Central Asia after around 60-40 ka (Callaway, 2015), for example, In Denisova Cave,
the Altai Mountains, Russia. The phalanx was found in a stratum dated to 48–30 ka ago (Krause et
al., 2010) (Figure 8). So, it is not difficult to link the local fire anomalies during 47.5-36 ka in the
Ili Basin to human activities: the increased occurrence of local fires (for cooking, or burning the
uncultivated land) quickly destroyed the vegetation, causing the observed vegetation degeneration.
If this is the case, the modern vegetation may have originated since around 36 ka. In future the use
of a massive and sustained ecological program of vegetation rehabilitation in the arid and semiarid
region should reduce the risk of destructive fire in order to avoid a similar local vegetation disaster
to that which occurred at 36 ka.

**5. Conclusions**

In the Nileke Section, Ili Basin, the pollen assemblages show a sharp change at ~36 ka

characterized by herb increase and Cupressaceae decrease, which is difficult to be explained in
terms of a Eurasian climate anomaly and instead is attributed to local vegetation degradation
caused by local fire intensification. Human activities during 45-36 ka are inferred as the main
driving force of this change, although direct archeological proofs are still lacking. In future, new
archeological sites in this area are required to investigate the extent to which ancient human
activities influenced the vegetation. This will provide further insights into the relationships
between human fire activity and local vegetation and even climate change.

**Acknowledgements**

The project is supported by the Natural Science Foundation of China (Nos: 41572162,

41472147 and 41271215), the National Key Research and Development Program of China (Nos:
2016YFA0601902), International partnership Program of Chinese Academy of Science (grant
number: 132B61KYS20160002), and the State Key Laboratory of Loess and Quaternary Geology,
Institute of Earth Environment, CAS (SKLLQG1515) and Open Foundation of MOE Key
Laboratory of Western China's Environmental System, Lanzhou University (lzujbky-2015- bt01).
The authors thank Y. Li, X. Li, J. Dong and F. Zhang for sampling and laboratory assistance.



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
