# Peer review of "Discussion started: 20 April 2017 © Author(s) 2017. CC-BY 3.0 License."

_Climate of the Past, 2017_

## Referee Comment (RC1) · Anonymous Referee #1 · 7 Jun 2017

This manuscript presents a long-term vegetation and fire history from Central Asia covering the last ca 70k yrs. The authors combine pollen and charcoal data from a loess sequence and discuss the relative roles of climate and human impact for the observed ecosystem changes. On the base of their vegetation and fire data, and of other available paleoclimatic sequences, they conclude that human impact is the most likely factor, supporting the idea of an "early human impact" on ecosystem changes in this region. The question whether humans may have modified vegetation and fire well before the current interglacial is certainly intriguing, and with many implications for our understanding of the legacy of human impact on pre-Holocene ecosystems. Unfortunately, neither evidence from their data, nor the argumentations presented in the discussion session, seem to support particularly this idea, which thus emerges as speculative and very elusive from their conclusions (see major points below). The paper should be refocused on a more balanced view of the possible causes of vegetation changes (including all possible factors, see below), and considering that the weight of evidence at the present does not fully support any prevailing cause. Major points 1) Anthropogenic drivers of ecosystem changes after ca 36k. The authors conclude that the lack of a marked climatic signal (as inferred from NGRIP, and other local-to-regional archives,) implies that the major driver of ecosystem change should be attributed to humans But as "absence of evidence is not evidence of absence", this argument appears rather weak, and not supported by data. The same vegetation/fire change after 36k could be explained by a series of other factors (which the authors seem not to fully account for) and in particular: 1) ecosystem response to a local climatic event, which is not evident at regional to continental scale. In this respect, the authors do not have any climatic proxy in their data, and thus their discussion of climate variability is quite limited. 2) a taphonomic effect due to loess deposition/changing sedimentological conditions, particularly for charcoal deposition (see also point 2). In addition, more direct evidence of humans in the area (e.g. from archeological data, or from direct anthropogenic pollen indicators) are lacking. Therefore, conclusions such as "it is not difficult to link the local fire anomalies during 47.5-36 ka in the Ili Basin to human activities: the increased occurrence of local fires (for cooking, or burning the uncultivated land) quickly destroyed the vegetation, causing the observed vegetation degeneration." (L 322-325) are not fully supported by the data. Similarly, a statement such as " the coeval local fire intensification supports human activity as a factor causing fire anomalies after around 6 ka. This relationship can be similarly extended to observed fire anomalies at 47.5-36 ka" (L315-418) seem very controversial considering that population size was remarkably different, and thus not directly comparable. 2) The authors analyze microscopic charcoal of various size classes, account for morphological structure, and link to specific

fire properties (e.g. frequency and severity/intensity). Unfortunately, such properties of the fire regime are very difficult to be discerned from microscopic charcoal alone, and lacking a calibration study specific for this archive (loess) and the location. Other factors (change in fuel type, depositional processes, what else?) may be also responsible for some of the observed patterns. Most importantly, charcoal is usually reported as influx, rather than concentrations (as it seems to be the case here), thus does not account for the changes in sediment rates evident from your depth/age relationship (Fig. 2).

3) The main conclusion that" In future, the use of a massive and sustained ecological program of vegetation rehabilitation in the arid and semiarid region should reduce the risk of destructive fire in order to avoid a similar local vegetation disaster to that which occurred at 36 ka" sounds quite anachronistic. I suggest rephrasing, or even remove it.

Minor points. - Introduction: first 2 paragraphs are too general and could be better focused, e.g. highlighting the lack of paleorecords in this deposition environments, and how loess can be an alternative archive. - L83. "...prevailing westerly winds, down its axis". Can you clarify? I'm not sure what you mean here. -L102 not clear what you mean for "rubification" in the figure. - L 155, and L213-214. You should better explain what "similar concentration and percentage pollen" do mean for the overall interpretation of the pollen record. - L179 should be Asteraceae - L186. I do see changes in the charcoal, but I don't see a sharp change after 36 k, compared to before. - Fig 2. Not clear what the blue/green and red series do represent in the final depth-age model. - Fig 3. A better age scale would help. Plus, adding charcoal would make the charcoal/pollen comparison easier. - Fig 6. You state that "no (climate) anomalies occurred during 41-36k in the climate proxy presented in the figure. The "arid index" from Central Asia, though, seem to show a significant increase in aridity, which is almost synchronous with the fire/vegetation change that you discuss. What do you mean, then, by "anomaly"? Also note that you should also account for the age

uncertainties among the different records, before attempting any regional comparison.

---

## Referee Comment (RC2) · Anonymous Referee #2 · 12 Jun 2017

This manuscript focuses on vegetation changes and fire dynamics in Central Asia (Ili Basin) from 70 to 0 ka BP reconstructed by means of pollen/spores and microscopic charcoal fragments analysis of a 20 meters loess section. The final aim is to understand the driving factors of the presented changes considering both climate and human activity. Due to the geographic area, the time interval studied, and also to the fact that there was not a previous complete palynological record, this manuscript presents an important and interesting scientific topic. The structure of the manuscript is generally clear and references are adequate. The main issue is that the interpretations and

conclusions are only partly supported by the pollen and microcharcoal results. The conclusion that the main vegetation change occurring at 36 ka is to be attributed to increased local fire activity caused by human activity remains speculative at this point, without direct archaeological evidences. Specific questions that might help to re-focus the discussions (thus conclusions and the title) are: 1) are there any anthropogenic indicators present in the palynological records? 2) what is the reason for the increase in the sedimentation rates during the interval ∼47.5-41 ka (Fig.2)? More dust supply from the Westerlies? I do see an increase in the grain size in your Nileke section record. How does it influence the pollen and microcharcoal records? 3) You say that "no anomalies occurred during 41-36 ka" (line 238, Fig. 6 caption), but what is then the peak between ∼41-39 ka evident in your records (Fig.6)?. I do see indication of a slight increase of aridity in the Ice Core Gulia from the Tibetan Plateau after 36 ka, which is in agreement with the increased fire occurrence indicated by your records (Fig.6). Minor comments: In the caption of Figure 1 it is better to indicate the names of the wind systems shown with the arrows. Figures. 3 and 4 could be a bit bigger. Figure 6 must be bigger to allow an easier proxy comparison. It would be helpful to add an horizontal line indicating 47.5 ka (also if is not a CONISS division). Figure 5 needs a more complete caption with a bit more explanation..the reader will understand why curves are oranges /blu just in the discussion. The paragraph of lines 218- 226 is very important to understand all the discussion, probably you can move it before, in section 4. Technical corrections: 1) "-e" is missing in "Nilke" line 90, caption Fig.1, 2)line 179, Asteracea"-e" is missing. 3) line 190, caption Fig. 5, R:... "round" is missing. 4) a verb is missing in the sentence from line 255 and line 257 (section 4.2). 5) within references: something is probably missing at reference "Conard NJ (2008)" line 383. 6) double reference for Song YG, Chen XL, Qian LB et al. 2014, Quaternary Int.

---

## Author Comment (AC2) · 10 Aug 2017

The comment was uploaded in the form of a supplement:
https://www.clim-past-discuss.net/cp-2017-62/cp-2017-62-AC2-supplement.zip

---

## Author Comment (AC3) · 10 Aug 2017

The comment was uploaded in the form of a supplement:
https://www.clim-past-discuss.net/cp-2017-62/cp-2017-62-AC3-supplement.zip

---

## Author Comment (AC5) · 3 Feb 2018

Dear Carlo, Thanks again for your further encouraging comments to revise our paper. Before response we have organized twice discussions within our group and then with Dr. D.J., Zhang (An archeologist, Lanzhou University: https://www.researchgate.net/profile/Dongju_Zhang2), respectively. Now we began to totally understand your worries and tried to organize the manuscript better. In this re-newed manuscript, besides the first-hand pollen and microcharcoals within the same profile, we still can't help thinking another important attribution is to investigate the internal relationships between vegetation anomalies and special local fire. Although the conclusion seems a little novel, it should be relatively more reasonable speculative so far. Here, we show a carton model below to help understanding the explanation easily. In this renewed manuscript, we rephrased 'Abstract' greatly and 'Discussion' moderately in order to explain more clearly the relationships between the fire anomalies at ∼47.5-36.0 ka and vegetation anomalies at ∼36.0 ka. Figure 11 was renewed through deleting the human migration routes in order to avoid the debates in the archeologists. Every modification in this manuscript is shown as "Marked manuscript" for easy identification of changes. Then the final one is show as "Clean manuscript". Thank you again for re-considering this paper.

Sincerely yours, Yunfa Miao, Yougui Song, Yue Li, Shengli Yang, Yun Li, Yongtao Zhao

Please also note the supplement to this comment:
https://www.clim-past-discuss.net/cp-2017-62/cp-2017-62-AC5-supplement.pdf

―――――――――――――――――

**Natural condition**

**Human activities**

**Fig. 1.** Cartoons to show two end-member modelings for natural and human-related fires

**Supplement:**

There are three parts here.

- 1. Response to Editor Decision
  - 2. Marked manuscript
    - 3. Clean manuscript

Dr. Yunfa Miao Donggang West Road 320 Lanzhou, Gansu 730000 China

Feb. 03, 2018

Dear Carlo,

Thanks again for your further encouraging comments to revise our paper.

Before response we have organized twice discussions within our group and then with Dr. D.J., Zhang (An archeologist, Lanzhou University: https://www.researchgate.net/profile/Dongju Zhang2), respectively. Now we began to totally understand your worries and tried to organize the manuscript better.

In this renewed manuscript, besides the first-hand pollen and microcharcoals within the same profile, we still can't help thinking another important attribution is to investigate the internal relationships between vegetation anomalies and special local fire. Although the conclusion seems a little novel, it should be relatively more reasonable speculative so far. Here, we show a carton model below to help understanding the explanation easily.

In this renewed manuscript, we rephrased 'Abstract' greatly and 'Discussion'

moderately in order to explain more clearly the relationships between the fire anomalies at ~47.5-36.0 ka and vegetation anomalies at ~36.0 ka. Figure 11 was renewed through deleting the human migration routes in order to avoid the debates in the archeologists. Every modification in this manuscript is shown as "Marked manuscript" for easy identification of changes. Then the final one is show as "Clean manuscript".

Thank you again for re-considering this paper.

Sincerely yours, Yunfa Miao Yougui Song Yue Li Shengli Yang Yun Li Yongtao Zhao

| 1  | Vegetation and fire anomalies during the last ~70 ka in the Ili Basin, Central Asia                                                                                     |
|----|-------------------------------------------------------------------------------------------------------------------------------------------------------------------------|
| 2  |                                                                                                                                                                         |
| 3  | Yunfa Miao a, b *, Yougui Song b, c *, Yue Li b, d , Shengli Yang e , Yun Li b , Yongtao Zhaoa |
| 4  |                                                                                                                                                                         |
| 5  | a. Key Laboratory of Desert and Desertification, Northwest Institute of Eco-Environment and                                                                             |
| 6  | Resources, Chinese Academy of Sciences, Lanzhou 730000, China                                                                                                           |
| 7  | b. State Key Laboratory of Loess and Quaternary Geology, Institute of Earth Environment,                                                                                |
| 8  | Chinese Academy of Sciences, Xi'an 710061, China                                                                                                                        |
| 9  | c. Research Center for Ecology and Environment of Central Asia, Chinese Academy of Sciences,                                                                            |
| 10 | Urumqi, 830011, China                                                                                                                                                   |
| 11 | d. College of Resources and Environment, University of Chinese Academy of Sciences, Beijing,                                                                            |
| 12 | 100049 China                                                                                                                                                            |
| 13 | e. Key Laboratory of Western China's Environmental Systems (Ministry of Education), College of                                                                          |
| 14 | Earth and Environmental Sciences, Lanzhou University, Lanzhou, 730000, China                                                                                            |
| 15 | Corresponding author: miaoyunfa@lzb.ac.cn; ygsong@loess.llqg.ac.cn                                                                                                      |
| 16 |                                                                                                                                                                         |
| 17 | Abstract: Last glacial period Records of vegetation characteristics and fire activity obtained from                                                                     |
| 18 | the same profile can offer an opportunity to better understand paleoelimatic and paleoecological                                                                        |
| 19 | changes and their underlying driving forces. Here, we present sporopollen (spores and pollen) and                                                                       |
| 20 | microcharcoal data collected together from the wind-blown loess Nileke (NLK) section,                                                                                   |
| 21 | coveringrepresenting the last ~70 thousand years (ka) in the Ili Basin (Northwest China), Central                                                                       |
| 22 | Asia. We found Results reveal that the temperate woody taxa (e.g., Cupressaceae) remained at high                                                                       |
| 23 | levels before 36.0 ka, while the total microcharcoal concentrations (MC) were relatively low.                                                                           |
| 24 | Aafter that 36 ka, they decreased and were replaced by herbaceous taxa (e.g., Artemisia,                                                                                |
| 25 | Chenopodiaceae). Such results indicate a special, localized ecological deterioration event (EDE)                                                                        |
| 26 | because it is hard to be explained in terms of a hemispheric-scale phenomenon; While the total                                                                          |
| 27 | microcharcoal concentrations (MC) show no obvious changes an increase at that timeabruptly                                                                              |
| 28 | replaced the woody taxa and the MC increased. This Such results indicate a notable vegetation                                                                           |
| 29 | degeneration and more fire (frequency/severity) at 36 kavegetation degeneration at 36 ka is                                                                             |
| 30 | notable because no equivalent changes have been identified anywhere else across Eurasia., which                                                                         |

| 31 | is hard to be explained in terms of a hemispheric scale phenomenon rather than a special,           |
|----|-----------------------------------------------------------------------------------------------------|
| 32 | localized ecological deterioration event (EDE); A, meaning no obvious fire frequency/severity       |
| 33 | happened; whereasnother interesting observation is that the EDE the vegetation degeneration         |
| 34 | immediately followed a period of intensified local fire characterized by an increased number of     |
| 35 | larger microcharcoal particles, in contrast to the smaller sizes occurring between 47.5 and 36.0 ka |
| 36 | characterized by an increased number of larger microcharcoal particles, in contrast to the smaller  |
| 37 | sizes, is immediately followed by EDE. No reasons can be well used to explain such unique           |
| 38 | results, neither the special taphonomic effects nor sedimentary processes. S. This pattern can be   |
| 39 | explained in terms of (1) a special, localized environment event caused by the particular special   |
| 40 | taphonomic effects or sedimentary processes unrelated to the fire strength/frequency; or (2) an     |
| 41 | ecological event driven by human activities, such as burning the local vegetation near the NLK      |
| 42 | site. The ynchronous latter case is argued to be more likely due to its coincidence of timing with  |
| 43 | the archaeological and fossil hominin sites discovered in the Central Asia meant that the humans    |
| 44 | have colonized the Ili Basin, who might have produced such local fire and eventually lead to the    |
| 45 | EDE through destructing local vegetation Our findings may provide new directions                    |
| 46 | opportunities-to assess the hypothetical mechanismrelationship among climate, vegetation and        |
| 47 | human activities: the human colonization into Ili Basin might have destroyed the local vegetation   |
| 48 | through burning and eventually lead to the EDE. F, and Ffuture analysis of first-hand               |
| 49 | archeological sitesproofs in the Ili Basin will be an important key step in checking this           |
| 50 | hypothesisargumenthypothesis                                                                        |

52 **Keywords**: Vegetation; Fire; Anomaly; Human activities; Last glacial period

53

**54 1. Introduction**

The climate, vegetation, fire and human activities, as well as the relationships among them during the late Quaternary, especially the last glacial period, provide basic insights into the interactions between the terrestrial ecosystem and the climate changes inby which to understand the future (e.g., Behling and Safford, 2010; Cheng et al., 2012; Li et al., 2013; Hubau et al., 2015; Varela et al., 2015). High-resolution stalagmite (Wang et al., 2001; Cheng et al., 2012), ice core (Thompson et al., 1997; Petit et al., 1999; Augustin et al., 2004) and loess (e.g., Chen et al., 1997;

| 61 | Hao et al., 2012; Sun et al., 2012; Rao et al., 2013) analyseis haves yielded many paleoclimate                       |
|----|-----------------------------------------------------------------------------------------------------------------------|
| 62 | records since the last glacial period, which . These are characteristic characterized by a series of                  |
| 63 | rapid and hemisphericallyhemispherically asynchronous climate oscillationsstrong fluctuations,                        |
| 64 | named cold Heinrich or warm Dansgaard-Oeschger events, as well as a warm middle Holocene                              |
| 65 | (e.g., Bond et al., 1997). However, there are uncertainties in the timing and amplitude of the                        |
| 66 | response of the terrestrial ecological landscape to the climate changes as the most sensitive organic                 |
| 67 | proxies for terrestrial climate change, a limited number of complete vegetation records have been                     |
| 68 | obtained to show how the terrestrial ecological landscape responded to the climate change, due to                     |
| 69 | the limited number of complete vegetation records (e.g., Guiot et al., 1993; Allen et al., 1999;                      |
| 70 | Jiang et al., 2011; Nigst et al., 2014). Previous studies These have revealed that the vegetation                     |
| 71 | changes are largely a response to natural climate change, with no strong evidence to suggest that                     |
| 72 | humans have significantly disturbed/changed the vegetation/ecology until the late Holocene (e.g.,                     |
| 73 | Nigst et al., 2014). Additionally, fFire is also a nother_another_sensitive proxy used for the                        |
| 74 | reconstruction of reconstructing climate and ecology (e.g., Filion, 1984; Bird and Cali, 1998;                        |
| 75 | Bowman et al., 2009). Besides climate and ecology, records of vegetation and fire together are                        |
| 76 | also unique indicators of _ as well as human activities, owing to the impact of human activities                      |
| 77 | such as vegetation cutting and burning (e.g., Patterson et al., 1987; Whitlock and Larsen, 2002;                      |
| 78 | Huang et al., 2006; Aranbarri et al., 2014; Miao et al., 2016a, 2017; Sirocko et al., 2016); however,                 |
| 79 | observational data in the arid areas of Central Asia span only several decades, and most relevant                     |
| 80 | studies have been limited to the late Holocene, especially at or near archeological sites, much                       |
| 81 | shorter than the natural fire rotation in these regions (Miao et al., 2017), considering the although                 |
| 82 | anthropogenic fire has been evidenced earlier than 1.0 million years 000 ka ago (e.g., Clark and                      |
| 83 | Harris, 1985; Gowlett and Wrangham, 2013). In fact, the later Pleistoceneast glacial period is                        |
| 84 | considered as a key period of modern human's migration: the human migration from Africa started                       |
| 85 | at ~200 ka to ~300 ka agoand spread into Eurasia (Templeton, 2002; Sun et al., 2012;                                  |
| 86 | Schlebusch et al., 2017), effects of human activities on the regional environments are ambiguous,                     |
| 87 | because vast majority of conventional results were acquired through comparisons of different                          |
| 88 | proxy records from different materials, these materials are always with different resolution and                      |
| 89 | dating uncertainties. Sso studies of vegetation and fire from a single within the same profile                        |
| 90 | (section or core) are helpful_uniquely helpful in understanding the vegetation, fire and climate $\frac{1}{2}$ |

- change, as well as human activities (e.g., Zhao et al., 2010; Wang et al., 2013; Miao et al., 2016a;
- 92 2017).